# From Perfect AUC to Poor Transfer: Diagnosing Leakage in Cross-Platform Gene Signature Learning

## ABSTRACT

Deep models often fail under distribution shift, yet the role of *feature selection* in amplifying or mitigating shift is underexplored. We study this in a stringent setting: transferring a tumour–vs–normal classifier across measurement *platforms* (Agilent microarray $\rightarrow$ RNA–Seq) using the same patients and genes. We introduce **SCOPES**, a leak–free, multi–objective selection framework that optimizes three competing goals: (i) predictive performance (AUC) via patient–safe cross–validation, (ii) selection stability (Kuncheva), and (iii) cross–platform alignment (Maximum Mean Discrepancy, MMD). On matched TCGA–BRCA Agilent/RNA–Seq, a label–informed $F$–score slab produced an implausibly perfect source model (AUC $\approx 1.0$) but lost $\sim 0.30$ AUC after transfer, revealing selection leakage plus platform shift. Replacing the slab with an *unsupervised* MAD prefilter makes the trade–off explicit on the Pareto front: a one–gene, alignment–first solution achieves modest AUC with small transfer loss ($0.69 \rightarrow 0.61$, $\Delta\text{AUC} \approx -0.08$), while a 30–gene, accuracy–first solution reaches near–perfect source AUC but transfers poorly ($\Delta\text{AUC} \approx -0.38$). SCOPES provides a simple protocol to *measure* and *control* this trade–off (report source/target AUC, $\Delta\text{AUC}$, and MMD), encouraging selections near a Pareto "knee" for portability. Finally, in the reverse direction (RNA–Seq$\rightarrow$microarray), a 37-gene SCOPES signature attains $\text{AUC}_{\text{RNA}} = 0.654$ (CV) and $\text{AUC}_{\text{Agilent}} = 0.890$ ($\Delta\text{AUC} = +0.236$), indicating directional shift. We argue that treating selection as a multi–objective design problem is a useful lens for the science of deep learning under shift.

## 1 INTRODUCTION

Understanding *why* models generalize—and when they fail—is central to the science of deep learning. Distribution shift exposes these limits: models that score highly on the source distribution can collapse on a shifted target. While representation learning and domain adaptation are widely studied, the upstream step of *which features we allow the model to see* can itself create or reduce shift. In biomedicine, this is stark for gene expression: legacy *microarrays* and modern *RNA–Seq* measure the same biology with different noise models and dynamic ranges, so selection that looks optimal on one platform may not transfer to the other.

We study cross–platform generalization on matched TCGA–BRCA patients measured by Agilent 244K microarrays and Illumina RNA–Seq V2. A naive, label–informed prefilter (global $F$–score on the full cohort) appears to "solve" the task on microarrays (AUC $\approx 1.0$) yet loses $\sim 0.30$ AUC when evaluated on RNA–Seq for the *same* patients—a textbook mixture of selection leakage and platform shift. This motivates a leak–free pipeline and an objective that explicitly balances accuracy with portability.

We propose **SCOPES** (Stability– and Cross–Platform–Optimized Expression Selection): a multi–objective framework that optimizes three quantities in tension—(i) patient–safe CV AUC on the source platform, (ii) selection *stability* across resamples (Kuncheva index), and (iii) cross–platform *alignment* measured by Maximum Mean Discrepancy (MMD). Conceptually, this mirrors a multi–head loss, but we solve it via Pareto optimization (NSGA–II) to expose the full trade–off surface rather than bake in weights. To prevent leakage while keeping the search tractable, we replace the

label–informed slab with an *unsupervised* variability filter (robust MAD across platforms), and we enforce patient integrity via StratifiedGroupKFold. A fuller discussion of cross-platform normalization and single-platform gene selection pipelines is provided in Appendix A.

**Contributions.** **(1)** A leak–free selection protocol that separates unsupervised slab construction (MAD) from downstream multi–objective optimization, with patient–safe CV. **(2)** A principled triad of objectives—AUC, stability, and MMD—that makes the accuracy–vs–portability tension explicit on the Pareto front, along with a simple reporting rubric (source/target AUC, $\Delta$AUC, MMD). **(3)** An empirical study on matched Agilent/RNA–Seq BRCA showing that larger panels can buy source AUC at the expense of transfer, while tiny panels reduce $\Delta$AUC but cap absolute performance; selecting near a Pareto "knee" offers a practical compromise. **(4)** A perspective for deep learning: *treat feature selection itself as an objective design problem under shift*, not a fixed preprocessing step, to better interrogate generalization in deep pipelines.

## 2 METHODS

### 2.1 MATCHED TCGA COHORT

We study cross-platform generalization using TCGA–BRCA patients measured on two platforms: Agilent microarray (source) and Illumina HiSeq RNA–Seq V2 (target). We keep only primary tumor (code 01) and solid tissue normal (code 11). For each patient and platform, we retain a single aliquot (lowest portion/vial code) to prevent correlated replicates leaking across validation folds. We then intersect patients and genes across platforms and re-index both matrices into an identical order, yielding paired matrices $\mathbf{X}^{\mathrm{arr}}, \mathbf{X}^{\mathrm{rna}} \in \mathbb{R}^{n \times p}$ with $n = 530$ and $p = 16{,}146$. Labels $\mathbf{y} \in \{0,1\}^n$ indicate tumor (1) vs. normal (0), and groups $g_i$ are patient IDs (first 12 characters of the TCGA barcode) used for patient-safe cross-validation.

### 2.2 PREPROCESSING AND SEARCH-SPACE REDUCTION

We remove genes with $\geq 50\%$ missing values on either platform, and impute remaining missing entries using per-gene medians computed on the microarray platform and applied to both platforms (to avoid platform-specific imputation shifts). Because evolutionary subset search over all genes is intractable, we restrict optimization to a *slab* of $p_0 = 1000$ genes. We consider two slab strategies: (i) a label-informed ANOVA $F$-score ranking on microarrays (useful but leakage-prone if computed globally), and (ii) a *leak-free* unsupervised variability slab using robust median absolute deviation (MAD) aggregated across both platforms (details in Appendix B.1).

### 2.3 SCOPES: MULTI-OBJECTIVE SUBSET SELECTION VIA NSGA-II

On the $p_0$-gene slab, SCOPES searches binary masks $\mathbf{s} \in \{0,1\}^{p_0}$ with a size constraint $\|\mathbf{s}\|_0 \leq k_{\max}$ (we use $k_{\max} = 120$). Each candidate subset is evaluated by three objectives that capture competing desiderata under shift:

1. **Source predictive performance** on microarray: $f_1(\mathbf{s}) = 1 - \mathrm{AUC}_{\mathrm{cv}}(\mathbf{s})$, where $\mathrm{AUC}_{\mathrm{cv}}$ is computed with StratifiedGroupKFold to ensure patient integrity.

2. **Selection stability** across resamples: $f_2(\mathbf{s}) = 1 - \mathrm{Kun}(\mathbf{s})$, where Kun is the averaged Kuncheva index over resampling splits (Appendix B.3).

3. **Cross-platform alignment** between the paired feature distributions: $f_3(\mathbf{s}) = \mathrm{MMD}_\gamma(\mathbf{X}_{\mathbf{s}}^{\mathrm{arr}}, \mathbf{X}_{\mathbf{s}}^{\mathrm{rna}})$, computed with an RBF kernel after per-platform z-scoring (Appendix B.4).

We minimize $(f_1, f_2, f_3)$ using NSGA-II, which returns a Pareto set of non-dominated trade-offs rather than a single weighted solution.

### 2.4 MODEL SELECTION AND EVALUATION

From the Pareto set, we report two deterministic selection rules to illustrate different scientific choices: (A) **alignment-first** pick: lowest MMD with AUC tie-break; (B) **size/stability** pick: en-

force a size window and minimum stability, then choose the best AUC among low-MMD candidates (full rule in Appendix B.6). For each chosen subset, we report (i) within-source AUC using patient-safe CV on microarrays and (ii) cross-platform AUC by training on the full microarray data and evaluating on RNA–Seq for the same matched patients. Transfer is summarized by $\Delta\text{AUC} = \text{AUC}_{\text{target}} - \text{AUC}_{\text{source}}$.

## 3 RESULTS

We report within-source discrimination on Agilent microarrays ("source") and cross-platform discrimination on RNA–Seq ("target"). Transfer is summarized by $\Delta\text{AUC} = \text{AUC}_{\text{target}} - \text{AUC}_{\text{source}}$; more negative values indicate larger platform shift. We also report alignment by $\widehat{\text{MMD}}_\gamma$ (smaller is better) and selection stability by mean Kuncheva (larger is better).

**A leakage-prone baseline looks perfect on-source but drops after transfer.** With a label-informed $F$-score slab computed globally (i.e., before cross-validation), NSGA-II selected a $k{=}105$ gene subset that achieved an apparently perfect source performance ($\text{AUC}_{\text{source}} \approx 1.00$) and near-perfect stability. However, when the model trained on microarrays was applied to RNA–Seq measurements for the same matched patients, performance fell to $\text{AUC}_{\text{target}} \approx 0.701$ ($\Delta\text{AUC} \approx -0.30$). This pattern is consistent with (i) feature-selection leakage inflating cross-validated source AUC and (ii) residual platform shift affecting portability. We therefore treat this result as an *over-optimistic* reference point rather than a reliable signature.

**Leak-free slab reveals a clear accuracy–alignment trade-off.** To remove label leakage in slab construction, we replaced the $F$-score slab with an unsupervised robust-MAD slab and re-ran NSGA-II with the same objectives (AUC, stability, MMD). Figure 1a shows the resulting Pareto set: solutions with higher source AUC tend to incur worse alignment (higher MMD), while low-MMD solutions often have lower source AUC. From this set we illustrate two deterministic picks representing different priorities:

**Run A (alignment-first):** selecting the minimum-MMD solution yields a single-gene subset ($k{=}1$) with modest source AUC but relatively small transfer degradation: $\text{AUC}_{\text{source}} \approx 0.69$, $\text{AUC}_{\text{target}} \approx 0.61$, $\Delta\text{AUC} \approx -0.08$.

**Run B (accuracy-biased):** enforcing a larger subset (via a size/stability filter) and selecting among low-MMD candidates by best source AUC yields $k{=}30$ genes with near-perfect source discrimination ($\text{AUC}_{\text{source}} \in [0.996, 1.000]$) but substantially worse transfer: $\text{AUC}_{\text{target}} \approx 0.615$ and $\Delta\text{AUC} \approx -0.38$.

Figure 1b summarizes source and target AUCs for the leakage-prone baseline and the two leak-free selections. Together, these results highlight the central finding: *more genes can buy source accuracy at the expense of portability*, and the Pareto front makes this trade-off explicit.

**Reverse transfer (RNA–Seq $\rightarrow$ Agilent) shows asymmetric shift.** To further test portability, we evaluated the same leak-free SCOPES selection under *reverse* transfer: we trained on RNA–Seq and evaluated on microarrays for the same matched patients. Using a $k{=}37$ subset (MAD slab; stability-aware SCOPES), the within-source RNA–Seq CV performance was $\text{AUC}_{\text{RNA}} = 0.654$, while testing on Agilent yielded $\text{AUC}_{\text{Agilent}} = 0.890$ ($\Delta\text{AUC}_{\text{arr}\leftarrow\text{rna}} = +0.236$). This positive $\Delta\text{AUC}$ indicates that the platform shift is *directional* rather than symmetric: a signature that is only moderately predictive on RNA–Seq can still separate classes strongly on microarrays. We report this reverse-direction result as an additional, leakage-free extension that complements the Agilent$\rightarrow$RNA–Seq findings and supports evaluating portability in both directions.

**Additional analyses (Appendix).** We provide a transfer–alignment visualization (AUC vs. MMD), per-gene cross-platform agreement plots for the 30-gene subset, and evolutionary optimization diagnostics in Appendix C.

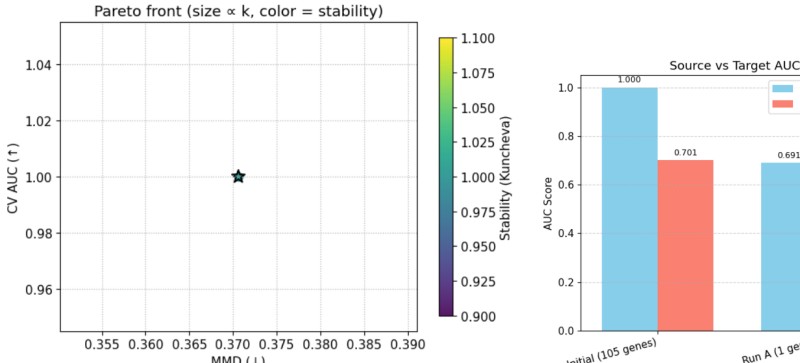

(a) **Pareto front (MAD slab).** Each point is a gene subset found by NSGA-II. x-axis: alignment ($\mathrm{MMD}_\gamma$, lower is better); y-axis: source CV AUC on Agilent (higher is better). Marker size $\propto k$; color encodes Kuncheva stability. Stars: Run A (min MMD) and Run B (accuracy-biased).

(b) **Source vs. target AUC.** Agilent (source) and RNA–Seq (target) AUC for baseline ($k{=}105$), Run A ($k{=}1$), and Run B ($k{=}30$). Baseline: high source AUC but transfer drop. Run A: smaller transfer loss at lower AUC. Run B: near-perfect source AUC but poor transfer.

Figure 1: **SCOPES trade-offs under leak-free slab construction.** (a) NSGA-II reveals an accuracy–alignment trade-off. (b) Selected operating points show that higher source AUC can coincide with larger transfer loss.

## 4    DISCUSSION

**Main finding.**    Cross-platform reuse of legacy microarrays with modern RNA–Seq is attractive, but platform shift makes naïve feature selection unreliable: a gene set can look highly predictive on the source platform while failing to transfer. Our experiments show this explicitly: the label-informed $F$-score slab produced near-perfect source AUC yet suffered a large transfer drop, whereas the leak-free MAD slab revealed a genuine trade-off between source accuracy and cross-platform alignment. Notably, reverse transfer (RNA–Seq→Agilent) shows a positive $\Delta$AUC, suggesting the mismatch is asymmetric; reporting both directions should be standard in cross-platform studies.

**Why multi-objective selection helps.**    SCOPES makes portability an explicit design goal by optimizing (i) source discrimination (AUC under patient-safe CV), (ii) stability (Kuncheva), and (iii) alignment (low $\mathrm{MMD}_\gamma$ between platforms on the chosen genes). This forces the optimizer to search for subsets that are not only predictive on microarray, but also *measured similarly* on RNA–Seq. As a result, the Pareto front exposes competing solutions: alignment-first subsets with smaller transfer loss versus accuracy-first subsets with larger transfer loss.

**Interpreting the two operating points.**    The two Pareto picks illustrate the trade-off: an alignment-first solution (very small subset) yields modest source AUC but relatively smaller $\Delta$AUC, while a larger subset can drive source AUC close to one yet transfer poorly, consistent with higher MMD. Practically, portable signatures should be selected near a Pareto "knee" where a small increase in subset size improves AUC without sharply worsening alignment.

**Limitations and Future Work.**    The BRCA matched cohort is imbalanced under our tumor/normal restriction (few normals), which can inflate separability and variance. Gene matching by symbol ignores probe/transcript design and may retain genes with systematically different measurement behavior. MMD depends on kernel scale (we use a standard heuristic). Transfer is evaluated only on matched patients; independent-cohort validation remains necessary for clinical claims. Despite these limitations, the observed pattern—source accuracy rising while portability degrades—is consistent with platform shift and selection bias, motivating reporting both $\Delta$AUC and an explicit alignment metric. Two immediate extensions can strengthen portability: (i) apply cross-platform calibration *before* selection and re-estimate the Pareto front; and (ii) use fully nested evaluation where slab and selection are recomputed within each fold. Replicating across additional TCGA cancer types and testing reverse transfer (RNA–Seq→microarray) would further assess robustness.

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

## A    APPENDIX: RELATED WORKS

Cross-platform transcriptomics has focused primarily on *distribution alignment* between microarrays and RNA–Seq, while feature selection is often treated as a separate, single-platform step. Feature-Specific Quantile Normalization (FSQN) and related calibration strategies can improve transfer and, when paired with iterative selection, approach within-platform performance Franks et al. (2018); Foltz et al. (2023); Skubleny et al. (2024). Beyond normalization, domain adaptation methods aim to learn domain-invariant predictors for cross-domain generalization; examples

include PRECISE for preclinical-to-tumor transfer and logit/KNN-based adaptation for robust classification under shift, though negative transfer can occur when assumptions are violated Mourragui et al. (2019); Yuan et al. (2023).

Within-platform gene selection remains highly active. Many pipelines combine a first-stage filter (e.g., variance or $F$-score) with a second-stage optimizer or hybrid strategy to obtain compact signatures with strong in-domain performance, including approaches based on meta-heuristics and group/cluster-guided selection Qu et al. (2021); Liu et al. (2023); Xu et al. (2024). Recent biomedical ML studies also emphasize multi-view/ensemble selection and sparsity-aware deep feature selection for building small predictive biomarker sets, as well as network-informed feature learning in single-cell or pan-cancer settings Chowdhury et al. (2025); Krishna et al. (2024); Kim & Jang (2024). Complementary work applies nested validation and staged filtering to identify compact validated biomarkers in other disease settings, highlighting the importance of evaluation protocol for preventing optimistic bias Thelagathoti et al. (2025); Tom et al. (2025).

Despite strong within-platform accuracy, portability is not guaranteed: genes that are highly discriminative on the source platform can be poorly aligned on the target platform. Our work targets this gap by treating portability as a first-class objective during selection through an explicit alignment term (MMD), while controlling selection bias via a leak-free prefilter and patient-safe evaluation.

# B  APPENDIX: EXTENDED METHODS

## B.1  SLAB CONSTRUCTION AND LEAKAGE

**Label-informed $F$-score slab (baseline).**  To shrink the feature space, we can rank genes on the microarray platform using a two-class ANOVA $F$-statistic and retain the top $p_0 = 1000$ genes. If this ranking is computed once using the full cohort, it leaks label information into cross-validation because each fold's test samples influence the selected features. This produces optimistic source AUC and stability. In the main text we use this baseline only to demonstrate leakage; our leak-free runs replace it with MAD.

**Leak-free robust MAD slab.**  For gene $j$ and platform $q \in \{\mathrm{arr}, \mathrm{rna}\}$, define the median $\tilde{\mu}_j^{(q)}$ and median absolute deviation $\mathrm{MAD}_j^{(q)} = \mathrm{median}_i |X_{ij}^{(q)} - \tilde{\mu}_j^{(q)}|$. We compute a platform-robust score by median-normalizing each platform's MAD and averaging:

$$\mathrm{rMAD}_j = \frac{1}{2} \left( \frac{\mathrm{MAD}_j^{(\mathrm{arr})}}{\mathrm{median}_g \, \mathrm{MAD}_g^{(\mathrm{arr})}} + \frac{\mathrm{MAD}_j^{(\mathrm{rna})}}{\mathrm{median}_g \, \mathrm{MAD}_g^{(\mathrm{rna})}} \right). \quad (1)$$

We retain the top $p_0 = 1000$ genes by $\mathrm{rMAD}_j$. This step uses no labels and is thus leak-free.

## B.2  PATIENT-SAFE CROSS-VALIDATION

We use StratifiedGroupKFold with groups as patient IDs to prevent a patient's samples from appearing in both train and validation. We report mean AUC across $K = 5$ folds.

## B.3  STABILITY METRIC (KUNCHEVA INDEX)

Let $S^{(a)}$ and $S^{(b)}$ be selected gene sets of equal size $k$ obtained under two resamples/splits. Kuncheva stability adjusts the overlap for chance under a universe size $p_0$:

$$\mathrm{Kun}(S^{(a)}, S^{(b)}) = \frac{|S^{(a)} \cap S^{(b)}| - \frac{k^2}{p_0}}{k - \frac{k^2}{p_0}} \in [-1, 1]. \quad (2)$$

We compute Kun across all split-pairs and average.

## B.4  ALIGNMENT OBJECTIVE (MMD)

For selected features, we z-score each gene within each platform to remove scale differences: $\tilde{X}_{ij}^{(q)} = (X_{ij}^{(q)} - \mu_j^{(q)})/\sigma_j^{(q)}$. We then compute $\mathrm{MMD}_\gamma^2$ with an RBF kernel $k_\gamma(\mathbf{u}, \mathbf{v}) = \exp(-\|\mathbf{u} -$

$\mathbf{v}\|_2^2/\gamma)$:

$$\text{MMD}_\gamma^2(\mathbf{X}, \mathbf{Z}) = \frac{1}{n_X(n_X-1)} \sum_{i \neq i'} k_\gamma(\mathbf{x}_i, \mathbf{x}_{i'}) + \frac{1}{n_Z(n_Z-1)} \sum_{j \neq j'} k_\gamma(\mathbf{z}_j, \mathbf{z}_{j'}) - \frac{2}{n_X n_Z} \sum_{i,j} k_\gamma(\mathbf{x}_i, \mathbf{z}_j).$$

(3)

We set $\gamma$ using the median pairwise-distance heuristic.

### B.5 NSGA-II DETAILS

We use NSGA-II with population size $M$ and $T$ generations. Masks are initialized to be feasible under $k_{\max}$, with crossover (HUX) and bit-flip mutation. A hard constraint $\|\mathbf{s}\|_0 \leq k_{\max}$ is enforced by rejection or constraint handling (as in code).

### B.6 PARETO SELECTION RULES

**Run A (alignment-first).** Choose the Pareto solution with minimum MMD; break ties by maximum CV AUC.

**Run B (size/stability constrained).** Filter Pareto solutions to a desired size interval and a minimum stability threshold; among remaining, choose the highest CV AUC, optionally restricting to the lowest MMD quantile.

### B.7 TRAINING AND EXTERNAL EVALUATION

For any chosen subset, we fit logistic regression on microarray features and report: (i) patient-safe CV AUC on microarrays and (ii) AUC on RNA–Seq by applying the same trained model to the paired RNA–Seq features for the same patients. We summarize transfer by $\Delta\text{AUC} = \text{AUC}_{\text{target}} - \text{AUC}_{\text{source}}$.

## C APPENDIX: ADDITIONAL RESULTS AND DIAGNOSTICS

### C.1 TRANSFER VS. ALIGNMENT VISUALIZATION

Figure 2 plots each candidate solution by $(\text{MMD}_\gamma, \Delta\text{AUC})$ with bubble size proportional to subset size. This makes the portability trade-off visually explicit: low-MMD subsets tend to show smaller transfer loss, whereas larger subsets can exhibit strong source performance but large negative $\Delta\text{AUC}$.

### C.2 PER-GENE PLATFORM AGREEMENT FOR THE 30-GENE SUBSET

Figures 3–4 show per-gene scatter plots of standardized expression (Agilent vs. RNA–Seq) for the same matched patients. Genes with clouds near the diagonal are better aligned; diffuse or nonlinear clouds indicate mismatch. This mixed alignment helps explain the higher $\text{MMD}_\gamma$ and poorer transfer of the 30-gene subset.

### C.3 EVOLUTIONARY OPTIMIZATION DIAGNOSTICS

We report (i) the number of non-dominated solutions per generation and (ii) the additive $\varepsilon$-indicator (symlog scale). These reflect search diversity and coarse progress rather than biological validity.

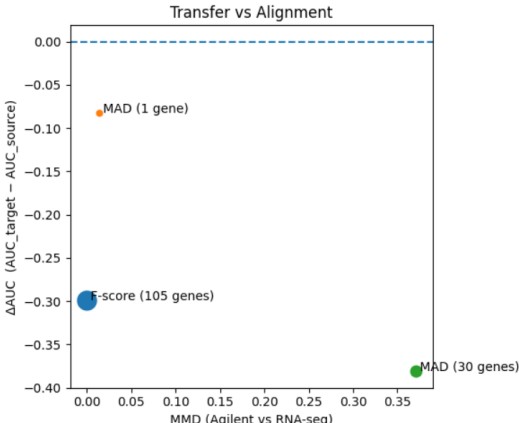

Figure 2: **Transfer vs. alignment.** Bubbles represent gene subsets positioned by $(\text{MMD}_\gamma, \Delta\text{AUC})$; bubble area is proportional to subset size $k$. The alignment-first solution (Run A) achieves low MMD and small transfer loss, while the larger subset (Run B) exhibits higher MMD and larger negative $\Delta\text{AUC}$.

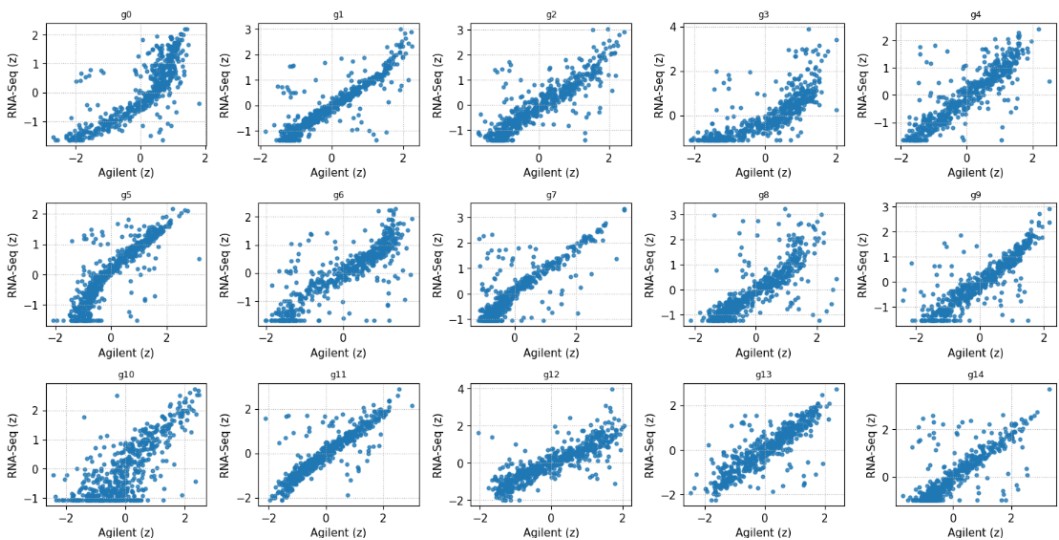

Figure 3: **Per-gene platform agreement (Run B, genes 0–14).** Each panel compares standardized expression on Agilent (x-axis) vs. RNA–Seq (y-axis) for the same patients. Panels near the diagonal indicate good agreement.

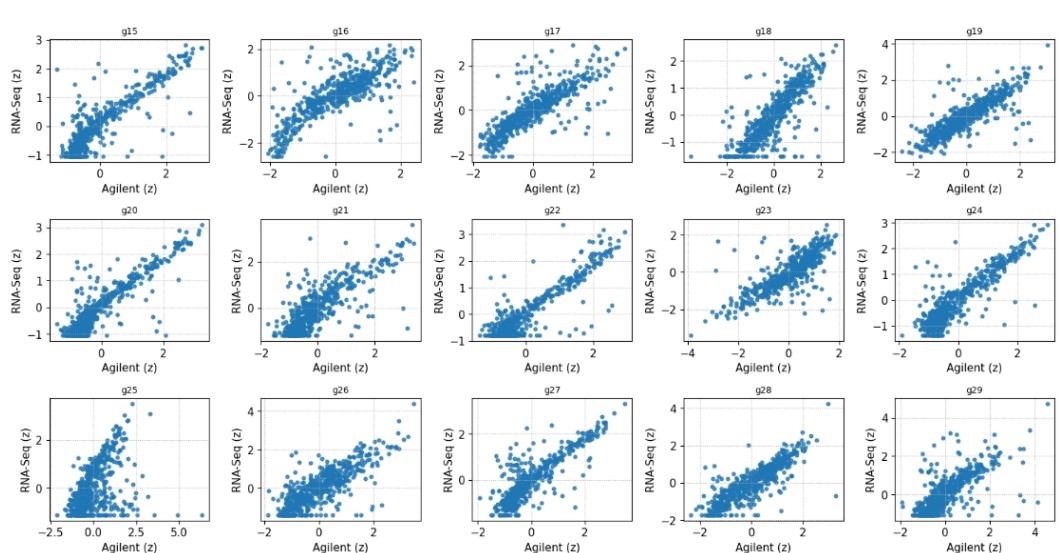

Figure 4: **Per-gene platform agreement (Run B, genes 15–29).** As in Fig. 3, alignment varies across genes, contributing to non-zero MMD and transfer loss.

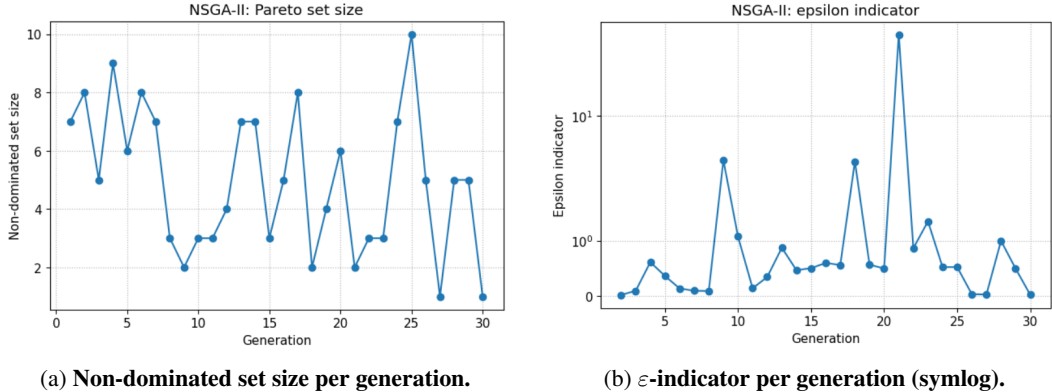

(a) **Non-dominated set size per generation.**  (b) $\varepsilon$**-indicator per generation (symlog).**

Figure 5: **NSGA-II diagnostics.** Diversity (left) and progress indicator (right) across generations.