# OpenReview forum: "From Perfect AUC to Poor Transfer: Diagnosing Leakage in Cross-Platform Gene Signature Learning"
_ICLR.cc/2026/Workshop/Sci4DL — Submitted to Sci4DL 2026_

### Official Review · Reviewer_YP7i · 2026-02-19

**Fit:** 1
**Significance:** 1
**Confidence:** 3

**Summary:**

This paper studies challenges with transferability of deep learning models from one gene expression measurement platform to another. They hypothesize that label-informed selection leads to overoptimistic AUC estimates on the source data due to selection leakage and that even once this issue is addressed by making selection unsupervised, there is a tradeoff whereby selecting more genes increases source AUC but also increases the transfer gap. They consider a framework for evaluating multiple objectives to understand the Pareto frontier of this tradeoff.

**Strengths:**

This paper considers an important problem of generalizability in deep learning, and grounds their study of it in a high-quality matched data set. Their analysis reveals large differences in approaches to this important problem that could inform future analyses.

**Suggestions:**

I'm concerned that this paper is not a good fit for the workshop. It seems very narrowly focused on a property more of the data than of deep learning itself, and the main contribution seems to be a method (which is again very specific to the application under consideration) for understanding these properties. Although they identify two distinct issues: selection leakage and distribution shift, it is unclear from the paper how much each contributes to the problem. Furthermore, there seems to me to be a third potential culprit, which is that one platform is simply noisier than the other, and this seems to be supported by the positive AUC transfer gap observed when the authors reverse the role of source and target. If this is the case, then the drops in AUC are not necessarily due to any failure of generalization--the lower AUC on the target could in fact just be the best that can be done on that data.

Another concern I have is that the issue of feature leakage is already very well-studied, and it is well-understood that one cannot select features based on the response and then expect CV to estimate performance accurately. In fact, there is also work showing that even unsupervised feature selection can bias CV's estimates (see https://academic.oup.com/jrsssb/article/84/4/1474/7073256) so it seems that even the authors' proposed fix may not necessarily be addressing feature linkage.

Minor point: I'm not sure what Fig 1a is supposed to show. I only see a single star in the middle of the plot, so nothing is really going on visually in this plot.

---

### Official Review · Reviewer_bguq · 2026-02-24

**Fit:** 2
**Significance:** 1
**Confidence:** 3

**Summary:**

This paper proposes a multi-objective gene selection framework for cross-platform transfer of tumour-vs-normal classifiers.

On a matched TCGA-BRCA microarray and RNA-Seq dataset, the authors show that a label-informed prefilter can introduce selection leakage, yielding near-perfect source AUC (~1.0) but a substantial drop (~0.30) after transfer. They replace this with an unsupervised MAD-based slab and optimize gene subsets jointly for source AUC, selection stability, and cross-platform alignment (MMD).

**Strengths:**

The paper raises a relevant and under-explored point: feature selection itself can amplify distribution shift. Framing selection as a multi-objective problem (accuracy, stability, alignment) is conceptually reasonable and clearly formulated.

**Suggestions:**

The paper is not easy to follow and the exposition can be improved. Besides the writing, several methodological gaps remain:

1) Limited generalizability.
All experiments are conducted on a single cancer type (BRCA) and a single platform pair. It is therefore difficult to assess whether the observed Pareto structure reflects a general phenomenon or dataset-specific artifacts. For example, the results may simply reflect that one modality is inherently more predictive (or less noisy) for this task and this explains the observed deltaAUC.

2) Pareto analysis is underdeveloped.
The central claim is about multi-objective trade-offs, yet conclusions are drawn from two chosen values of k. I would expect to see how all three objectives evolve as k varies. Without this, the “trade-off” remains anecdotal rather than fully characterized. Similarly, do we need all three objectives or a subset sufficient? Ablation studies feel natural in this setting!

Overall, while the framing can have potential, but the empirical evidence is too narrow to support the general claims about distribution shift made in this paper. There are simpler explanations that can explain the drop in predictive performance that the authors fail to address.

---

### Official Review · Reviewer_H8vy · 2026-02-27

**Fit:** 1
**Significance:** 2
**Confidence:** 2

**Summary:**

In this work, the authors investigate the role of feature selection in mitigating or amplifying distribution shift within a stringent genomics transfer learning setting (Agilent microarray to RNA-Seq).
They introduce SCOPES, a leak-free, multi-objective feature selection framework that optimizes for predictive performance (AUC), selection stability (Kuncheva), and cross-platform alignment (MMD).
The study empirically demonstrates that supervised selection prefilters suffer from severe leakage and transfer loss, whereas an unsupervised prefilter makes the accuracy-alignment trade-off explicit.

**Strengths:**

Detecting feature leakage in genome learning is a very important contribution towards better adoption of deep learning methods.

**Suggestions:**

The contribution is less towards a principled understanding of deep learning methods and more towards scientific methods in bioinformatics.

---

### Meta-Review · Area_Chair_tb7r · 2026-02-28

**Recommendation:** Reject

**Metareview:**

Reviewers gave low scores for significance and fit.

---

### Decision · Program_Chairs · 2026-03-02

Reject